# Response to Anthracnose in a Tarwi (*Lupinus mutabilis*) Collection Is Influenced by Anthocyanin Pigmentation

**DOI:** 10.3390/plants9050583

**Published:** 2020-05-02

**Authors:** Norberto Guilengue, João Neves-Martins, Pedro Talhinhas

**Affiliations:** 1DRAT, Instituto Superior de Agronomia, Universidade de Lisboa, 1349-017 Lisbon, Portugal; guilenguen@gmail.com (N.G.); nevesmartins@isa.ulisboa.pt (J.N.-M.); 2Agricultural Faculty, Agricultural Engineering Course, Instituto Superior Politécnico de Gaza, Lionde, 1204 Chókwè, Mozambique; 3LEAF, Linking Landscape, Environment, Agriculture and Food, Instituto Superior de Agronomia, Universidade de Lisboa, 1349-017 Lisbon, Portugal

**Keywords:** *Lupinus mutabilis*, *Lupinus albus*, *Colletotrichum lupini*, anthracnose, susceptibility, anthocyanin pigmentation

## Abstract

Anthracnose, caused by *Colletotrichum lupini*, is a major limiting factor for lupin production. Tarwi or Andean Lupin (*Lupinus mutabilis*) is generally regarded as susceptible to anthracnose, but the high protein and oil content of its seeds raise interest in promoting its cultivation in Europe. In this study we evaluated the response to anthracnose of 10 tarwi accessions contrasting in anthocyanin pigmentation, by comparison to white lupin (*Lupinus albus*), using a contemporary Portuguese fungal isolate. A severity rating scale was optimized, including weighted parameters considering the type of symptoms and organs affected. All tarwi accessions were classified as susceptible, exhibiting sporulating necroses on the main stem from seven days after inoculation. Anthracnose severity was lower on anthocyanin-rich tarwi plants, with accession LM34 standing out as the less susceptible. Accession I82 better combines anthracnose response and yield. In global terms, disease severity was lower on white lupin than on tarwi. Although based on a limited collection, the results of the study show the existence of genetic variability among *L. mutabilis* towards anthracnose response relatable with anthocyanin pigmentation, providing insights for more detailed and thorough characterization of tarwi resistance to anthracnose.

## 1. Introduction

Anthracnose, caused by *Colletotrichum lupini* (Bondar) Damm, P.F. Cannon & Crous, represents the most important disease in *Lupinus* and is known since the first half of the 20th century. It causes significant yield losses and is a major limiting factor for lupin cultivation, namely of lupin crops for seed production. Most *Colletotrichum* pathogens are polyphagous, with the same genetic entity found on multiple hosts. Moreover, frequently the same host is affected by multiple *Colletotrichum* spp., with no clear differences on the symptoms caused. However, the lupin anthracnose pathosystem seems to be an exception to this common trend, as lupin anthracnose is caused solely by *C. lupini* and *C. lupini* seems to prefer *Lupinus* spp. [1]. Additionally, very little genetic diversity is recognized among *C. lupini* populations, with only two groups reported: one corresponds to the North American outbreak in the first half of the 20th century, and currently not occurring in nature; the other corresponds to the contemporary outbreak, that began in the 1980s in Europe and is now found across the world [1]. Typical disease symptoms include twisting of stems, petioles and pods with necroses, in the centre of which arise acervuli. When ripe, acervuli produce conidia aggregates in mucilaginous masses of orange colour [2]. Tissues above the infection area may thus collapse, leading to plant death if the infection occurs on the main stem in early stages of the plant life cycle, compromising the crop.

The genus *Lupinus* (Fabaceae) comprises four domesticated species among over 280 species occurring in the American continent and around the Mediterranean Basin. White lupin (*L. albus* L.), yellow lupin (*L. luteus* L.) and narrow leaf lupin (*L. angustifolius* L.) are of Mediterranean origin, while tarwi or Andean lupin (*L. mutabilis* Sweet) is of South American origin. Lupins were domesticated for feed and food due to the high protein content of their seeds independently in the Mediterranean and Andean regions. During European colonization of America, tarwi remained mostly unnoticed as a neglected crop, but interest on it grew over the last few decades due to the high lipid and protein content of its seeds, and efforts are being made to introduce it to other parts of the world [3]. Besides the narrow genetic diversity typical of a recently domesticated species with no known wild specimens [4,5], the main challenges for selecting genotypes suitable for cultivation in different parts of the world are yield, yield stability and growth habit adaptable to the climatic conditions in each region [6]. Tarwi is generally regarded as susceptible to anthracnose [2] and anthracnose is considered the most important disease of this crop [7].

To ensure survival and continuity of the species some plants have developed self-defence mechanisms that confer ability to delay or prevent entry and/or development of the pathogen in the host [8]. Such mechanisms are based on physical barriers and biochemical responses. The biochemical responses have been the most studied and several phenolic compounds, including anthocyanin [9], are related with resistance against several pathogens. Anthocyanins are widely studied in many species of higher plants and have been associated with multiple biological functions such as fungitoxic, antibacterial and antiviral [10]. For instance, in coloured onions resistance against *Colletotrichum circinans* was reported due to the presence of catechol and protocatecoic acid toxic, unlike in white onions [11]. In corn it was noted that the accumulation of phenolic compounds, mainly flavones, reduced the size of the lesion conferring resistance to *Colletotrichum graminicola* [12]. Potatoes rich in anthocyanin presented resistance against *Pectobacterium carotovorum* [13]. Anthocyanin present in rice increased the resistance against the rice blast pathogen, *Magnaporthe grisea* [14]. In *Lupinus*, cases of resistance to *Colletotrichum lupini* were identified in some genotypes of *Lupinus angustifolius* and *L. albus* [2,15]. Thenceforth, these materials have been used as a source of resistance in breeding programs for these species.

As part of a research programme aiming to select tarwi genotypes adapted to cultivation under European conditions, the objective of this work is to characterize the response to anthracnose of *L. mutabilis* accessions by comparison to *L. albus*. A contemporary *C. lupini* strain was isolated from naturally occurring field infections in Portugal and characterized. This strain was used to characterize disease response in ten *L. mutabilis* accessions contrasting on their anthocyanin pigmentation.

## 2. Results

### 2.1. Isolation of Colletotrichum Lupini in Portugal

Sequencing results of a 968-bp fragment of *ApnMat1* gene for the two isolates (NG001, collected in Portugal, and RB221, collected in France, respectively with GenBank references MN783012 and MN783013) reveals 100% similarity between them. This indicates that the Portuguese isolate does not differ from the isolates currently occurring in the rest of the world, as isolate RB221 has been treated as representative of the contemporary lupin anthracnose outbreak.

### 2.2. Evaluation of Response to Anthracnose in Lupinus Mutabilis Accessions

Ten *Lupinus mutabilis* accessions were evaluated for anthracnose response using six *L. albus* accessions as reference [2]. All accessions were inoculated with *Colletotricum lupini* and the first symptoms appeared from seven days after inoculation. No accession was immune to the disease. Disease symptoms were observed on the stems, petioles and leaflets. Disease symptoms caused by *C. lupini* in *L. mutabilis* and *L. albus* accessions are presented in Figure 1. The area surrounding necroses on stems of accession LM18 developed anthocyanin pigmentation (consistently observed in all plants of this accession in both experiments; Figure 1c), while this did not occur in any other *L. mutabilis* accession nor in *L. albus* accessions. ANOVA performed for severity data reveal significant differences among accessions from seven days after inoculation onwards and also for the AUDPC, but not between both experiments (Table 1).

The Area Under Disease Progress Curve (AUDPC) was calculated for sixteen accessions as shown in Table 2. The AUDPC was calculated using trapezoidal rule from successive assessment data (seven, 14 and 21 days after inoculation). AUDPC in *Lupinus albus* accessions ranged from 0.79 (Bunheiro Murtosa) to 15.5 (Rio Maior), with an average of 7.8. AUDPC in *L. mutabilis* was higher than in *L. albus* accessions, ranging from 1.52 (LM34) to 19.2 (LM268) and the global average was 8.4. Among *L. mutabilis* accessions, there were significant differences between accessions with and without anthocyanin pigmentation (Table 3). *Lupinus mutabilis* accessions with anthocyanin pigmentation presented average AUDPC score of 3.25, while those with no anthocyanin pigmentation presented an average value of 16.2. The AUDPC scores of *L. mutabilis* accessions with anthocyanin pigmentation did not differ from those of *L. albus* accessions Bunheiro-Murtosa, Prima and MISAK, while the accessions with no anthocyanin pigmentation did not differ from *L. albus* accessions Lublanc, Estoril and Rio Maior.

## 3. Discussion

Anthracnose is a major constrain to sustainable lupin production worldwide and tarwi is generally recognized as susceptible to this disease. The objective of this work is to evaluate tarwi response to anthracnose under Mediterranean-climate conditions, as part of a broader initiative to select *Lupinus mutabilis* accessions adaptable to cultivation in Europe.

The lupin anthracnose pathogen occurring in Portugal had not been characterized for around 20 years [16]. *Colletotrichum lupini* isolate NG001, isolated from *L. mutabilis* in 2018 at Coimbra, Portugal, was shown to present 100% similarity in the *ApnMat1* to isolate RB221 from *L. albus* in Brittany, France, and considered as a representative strain of the current lupin anthracnose outbreak [1,17].

Our study demonstrated a severity differential response among accessions after the inoculation with *C. lupini*. According to our scale used to classify severity, several *L. mutabilis* accessions presented significantly lower disease levels that other accessions. Present results differ from previous studies [2,18,19], where *L. mutabilis* accessions were generally found as susceptible. These differences are clearly related with anthocyanin pigmentation, criterion used to define the accessions to be integrated in the assays. Anthocyanins can be found in many species under various forms, are known for their multiple biological functions such as fungitoxic, antibacterial and antiviral [10]. Several studies report the role of anthocyanins in the resistance to fungi of the genus *Colletotrichum* and other species in different cultures [12,13]. In mango fruit with anthocyanin and flavonoids accumulation was observed more resistance to a challenge of *Colletotrichum gloeosporioides* fungal inoculation and showed reduction in general decay incidence [20]. Weber et al. [21] identified individual phenolic compounds at different stages of *Colletotrichum simmondsii* infection in strawberry fruits and runners. Significant differences in individual phenolic compounds in strawberry fruits were detected at the beginning of the infection compared to uninfected fruits. These authors found a gradual increase in flavanols and anthocyanins with the progression of the infection, clearly showing the role of these compounds in disease resistance. Similarly, Lo et al. [22] studied the response of two sorghum cultivars (susceptible and resistant) to the interaction with *Colletotrichum sublineolum*. These authors found an incompatible interaction in the resistant cultivar having verified that the development of the fungus in the host was contained during the early stages of pathogenesis. They also noted greater and faster accumulation of phytoalexins including luteolinidine and 5-methoxyyluteolinidine, and an activation of defence-related genes such as the PR-10 protein. In the susceptible cultivar, they reported positive interaction, with colonization of the host and the proliferation of primary and secondary hyphae and without production of phytoalexins. These authors attribute the phytoalexins (3-deoxyanocyanidin) as the main component of resistance to *C. sublineolum* in sorghum. In corn with flavones was noticed resistance against *C. graminicola* [12]. Agrios [11] reported resistance against *C. circinans* in coloured onions. Biological roles of anthocyanins in other genera of fungi are also reported. Zhang et al. [23] verified that tomatoes with greater regulation of the genes involved in anthocyanin biosynthesis and anthocyanin accumulation better respond to the attack of gray mold. Schaefer et al. [24] evaluated the role of anthocyanins in the defence against *Botrytis cinerea*, *Mucor* cf. *racemosus*, *Sordaria* cf. *macrospora*, *Phoma herbarum*, *Mucor* sp., *Phoma* sp./*Didymella*, *Aureobasidium pullulans* and *Colletotrichum* sp. on blackberries and grapes. They found that there was a decreased risk of infection with *Botrytis cinerea* in grapes with anthocyanins. In experiments carried out on agar plates, growth inhibition of nine fungi that cause fruit rot in anthocyanin fruits was observed. In ripe blackberries the rate of growth reduction was 95%. In apple, anthocyanins were significantly increased in infected symptomatic tissue with *Gymnosporangium yamadai* [25]. All of these studies clearly demonstrate the role of anthocyanins in defence against various pathogens. In some species, the defence response occurs with disease progression. We found in our study that the accession LM18 without anthocyanins started the production of these compounds around to anthracnose-affected area (Figure 1c). These finding leads us to believe that the anthocyanins that occur in *L. mutabilis* have a role in fighting *C. lupini*. Several studies report that anthocyanins can be induced in plants in response to biotic or abiotic stress. We observed in this study that unlike accessions such as LM34, LIB222, I82, XM-5, LM13 and LM231 where anthocyanins occur naturally in the LM18 accession the anthocyanin induction was in response to the attack. We also noticed that a large part of the attacks occurred on the stems, mainly in the growth apices. This can be justified by the fact that the tissues are not lignified, which could have facilitated the growth and development of the fungus.

In our previous work, we mentioned that the accession LM268 was the one with the best yields under Mediterranean conditions [6], but it is highly susceptible according to the results of the present study. Therefore, there is a need to improve the resistance levels of this accession against anthracnose and the results of this work constitute an excellent starting point for the beginning of breeding programs. Based on the results of the two studies, we recommend the use of the LM268 accession in regions less prone to disease and for wet regions the use of the LM34 accession for having presented higher levels of resistance to disease. However, accession I82 is the one that better combines yield and low anthracnose severity.

## 4. Materials and Methods

### 4.1. Fungal Material

*Colletotrichum lupini* Isolate RB221 [17], isolated from *L. albus* in France in 2015, was used for comparison. The Portuguese isolate, named NG001, was obtained from infected tarwi stems and pods harvested in the experimental field of Estação Agrária (Coimbra) in 2018. Pieces of plant tissues on the rind of infections were surface-sterilized in NaClO 1% for 30 s, rinsed twice in sterile distilled water and placed in petri dishes containing Potato Dextrose Agar (PDA, Difco) medium amended with KSCN 0.5 mM, that were incubated at 25 °C in the dark. Conidia from *Colletotrichum*-looking colonies were used to generate a monosporic culture.

*Colletotrichum lupini* isolates were cultivated in PDA culture medium at 25 °C under 16 h of darkness and 8 h of light to stimulate spore formation for inoculation experiments. For fungal characterization, mycelium was cultivated in Potato Dextrose broth at 25 °C with occasional stirring. Mycelium was collected, filtered and freeze-dried.

### 4.2. Fungal Characterization

Fungal DNA was extracted from freeze-dried material using the DNeasy^®^ Plant mini kit (Qiagen, Germany). The extracted DNA was used for sequencing of the *ApnMat1* gene [26]. PCR was performed in the following conditions: pre-denaturation 5 min at 94 °C, 40 cycles of 1 min at 94 °C, 1 min at 55 °C, and 1 min at 72 °C and final extension at 72 °C for 20 min. The final volume was 25 µL, 8 µL of sterile H_2_O, 2.5 µL of forward and reverse primers, 12.5 µL of unstained dNTP + Taq DNA polymerase and 2 µL of DNA. The products were separated on 2% agarose gel on electrophoresis and, after purified, were sent for sequencing (StabVida, Portugal).

### 4.3. Plant Material and Growth Conditions

Before start with anthracnose resistance experiment, a trial was performed in 2017/2018 season using eleven accessions selected from our germplasm collection based on 12 phenotypic patterns expressed in the seed coat (Figure 2), following a completely randomized design with three repetitions. Plants were produced in vegetation boxes with 1 m^2^ and protected with an insect net. Each vegetation box contains six accessions with five plants each in a 20 × 20 cm spacing (Figure 3). This experiment was carried out to identify in the next generation anthocyanin pigmentation accessions (Figure 4), pointed for several biological roles in order to integrate in the anthracnose resistance experiment.

To study the resistance level in *L. mutabilis*, seeds of 10 accessions were selected (Table 4), considering also the results of phenotypic analysis and genetic characterization [6]. In parallel, previously analysed *L. albus* accessions (Prima and Bunheiro-Murtosa, Estoril and Misak, and Rio Maior and Lublanc, in increasing levels of anthracnose susceptibility) were used as reference [2]. All accessions were germinated in trays using thick river sand as substrate and grown inside Instituto Superior de Agronomia (ISA) greenhouse with temperature around 25 °C.

### 4.4. Inoculation and Evaluation of Disease Severity

From fungal material grown on PDA plates a spore suspension was obtained by flooding the plate using sterile water and with the aid of a loop fungus colony was scraped to separate the spores from the mycelium. The suspension was filtered using a number 1 mesh filter letting only the spores pass. Through the haemocytometer the spore concentration was adjusted to 2 × 10^6^ spores/cm^3^. To the spore suspension was added an equal volume of 2% gelatine as wetting agent bringing the final concentration to 1 × 10^6^ spores/cm^3^.

Plants with 7–8 leaves were inoculated with isolate RB221 using a manual spray and immediately placed in a humid chamber for 24 h at 25 °C in the dark. After 24 h were withdrawn from the humid chamber and returned to initial conditions cultivation. Experimental design adopted was completely randomized with three factors (experiment, accessions and days after inoculation), where for each accession was analysed 15 plants. Two anthracnose evaluation experiments were conducted. Severity disease was evaluated using the method proposed by Talhinhas et al. [2], with some improvements. Symptoms on the main stem were rated as: 0, no symptoms; 5, torsion without necrosis; 8, torsion with lateral necrosis; 10, torsion with necrosis totally surrounding the stem. This value was multiplied by the number of occurrences. Branches and flowers/pods were not evaluated as plants were inoculated and analysed prior to flowering and branch formation. Additionally, symptoms on petioles and leaflets were rated according to the type of symptom (weight factor of: 0.7 for torsion of petiole or leaflet without necrosis; 1.0 for torsion of petiole or leaflet with lateral necrosis; 1.3 for torsion with necrosis totally surrounding the petiole or leaflet) multiplied by the number of occurrences. A weighting factor of 0.2 was applied to the ‘leaflet’ score, so that five infected leaflets receive and identical score to on infected petiole. The final disease severity score ranges between 0 and 10. Calculated values above 10 were adjusted to 10.
S = n_si_ * S_i_ + n_Pi_ * f_i_ + 0.2 * n_Li_ * f_i_
S—disease severity;n_si_—number of occurrences of infection in the main stem;S_i_—severity in the main stem (0, no symptoms; 5, torsion without necrosis; 8, torsion with lateral necrosis; 10, torsion with necrosis totally surrounding the stem);n_Pi_—number of occurrences of infection in the petiole i;n_Li_—number of occurrences of infection in leaflet i;f_i_—weight factors (0.7 for torsion of petiole or leaflet without necrosis; 1.0 for torsion of petiole or leaflet with lateral necrosis; 1.3 for torsion with necrosis totally surrounding the petiole or leaflet).


The quantitative development and intensity of disease was measured by calculating the Area Under Disease Progress Curve (AUDPC) [27]. This parameter is widely important because it helps to categorize varieties under level of resistance. This technique is based on simple midpoint (trapezoidal) method that split disease progress curve in several trapezoidal determining individual area and finally adding the all areas. AUDPC determine disease progress along a time period and can be estimated using the following formula proposed by Madden et al. [28]. Accessions with AUDPC score below one were considered as resistant, while those with score values above five were considered susceptible. Accessions with scores between one and five were considered moderately susceptible.
$$\mathrm{AUDPC}=\overset{n-1}{\underset{i=1}{\sum }}\frac{\left(S_{i+1}+S_{i}\right)}{2}\;*\;\left(t_{i+1}-t_{i}\right)$$
where,
S_i_ = anthracnose disease severity on the ith date;t_i_ = date on which the disease was scored;n = numbers of dates on which disease was scored.


### 4.5. Statistical Analysis

Our data did not follow normal distribution and the variance was not homogeneous, therefore the results of severity and AUDPC obtained were compared individually among different accessions using ANOVA based on rank transformation used for non-parametric analysis [29]. Post-hoc Tukey HSD test of means was performed for two variables at 5% of significance. All analysis was performed in the RStudio program version 1.1.456 (The R consortium, Boston, MA, USA).

## 5. Conclusions

The present study suggests that the anthocyanins present in the stems of some *Lupinus mutabilis* accessions may play a role in containing the advance of *Colletotrichum lupini*. Our results highlight *L. mutabilis* accession LM34 rich in anthocyanins as being the least susceptible and LM268, without anthocyanins, the most susceptible. The LM268 accession has the particularity of being the most productive in our collection and thus the analysis of segregation in crosses between LM268 and LM34 may reveal genotypes high yielding genotypes with reduced susceptibility to anthracnose. Due to the higher production, accession LM268 can be recommended for regions less susceptible to disease and LM34 for wet regions. Accession I82 has the particularity of combining resistance and better yields. Although the majority of our *L. mutabilis* accessions with anthocyanins are classified as moderately susceptible, these accessions can serve as a source of resistance for the start of the breeding programme. The accessions of *L. albus* were less susceptible to the disease than tarwi accessions. The results of this study suggest a need for further exploration of anthocyanins including the identification of the different types of anthocyanins present in this species in order to better assist breeding programmes.

## Figures and Tables

**Figure 1 plants-09-00583-f001:**
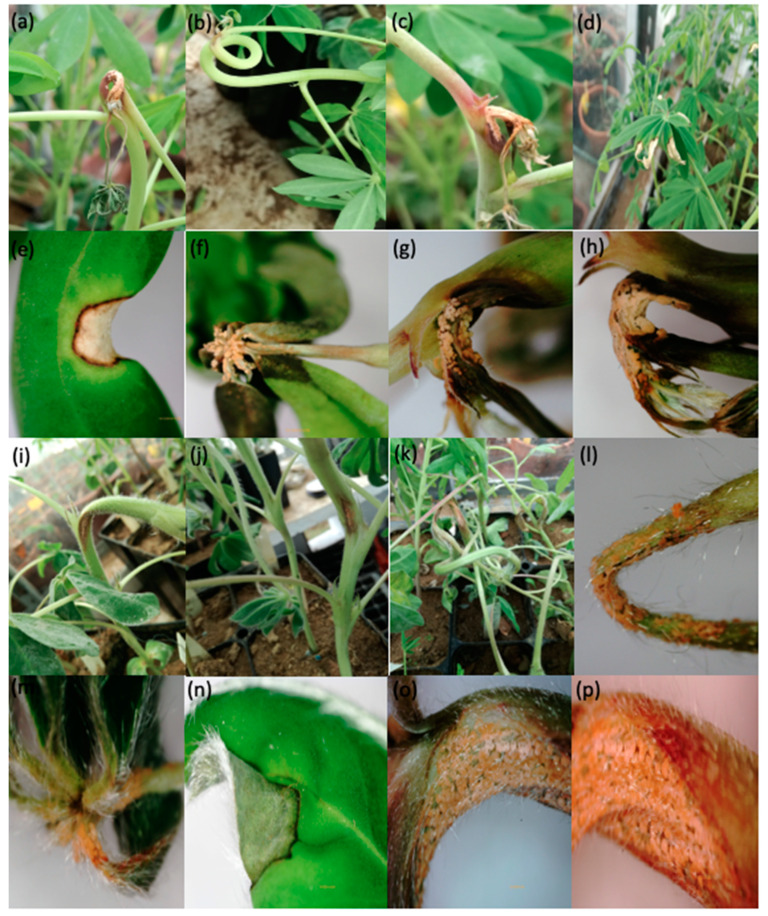
Anthracnose caused by *Colletotrichum lupini* on the main stem, petiole and leaflet of lupins. *Lupinus mutabilis*: (**a**–**c**) injuries in apical growth points of the main stem; (**d**–**f**) injuries on the leaflet and petiole; (**g**,**h**) production of conidia aggregates in mucilaginous masses of orange colour. *Lupinus albus*: (**i**–**k**) lesions on the main stem; (**l**–**n**) damage in petiole and leaflet; (**o**,**p**) production of conidia aggregates in mucilaginous masses of orange colour.

**Figure 2 plants-09-00583-f002:**
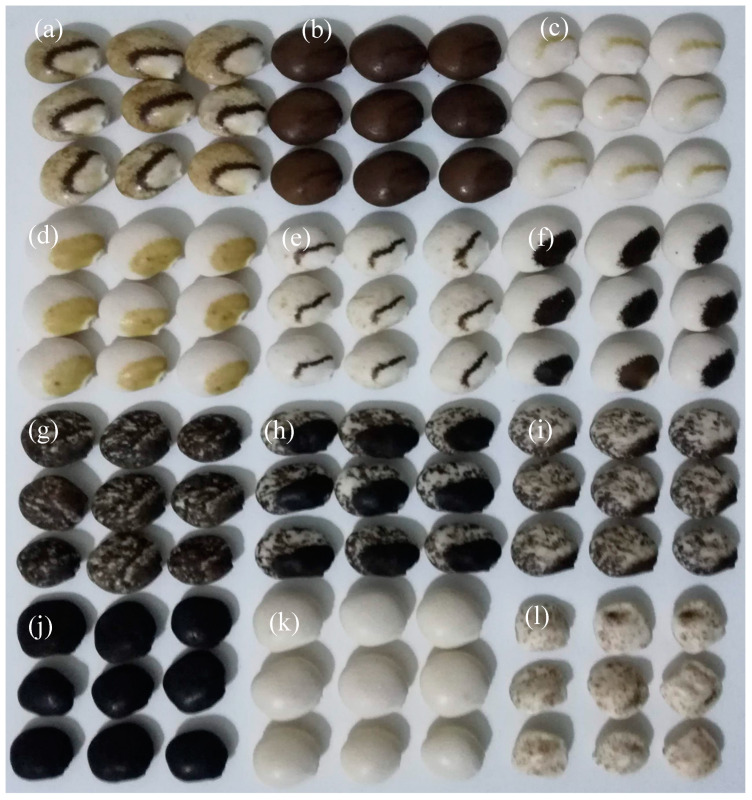
*Lupinus mutabilis* seed pattern pigmentation used for phenotype evaluation of plants and selection accessions with anthocyanin content. (**a**)-brown with over eye and specks; (**b**)-brown with over eye; (**c**,**e**)-white with over eye; (**d**)-white with crème crescent; (**f**)-white with brown crescent; (**g**)-white with moustache and specks; (**h**)-white with dark crescent and specks; (**i**)-white with moustache with specks; (**j**)-dark; (**k**)-white and (**l**)-white with specks.

**Figure 3 plants-09-00583-f003:**
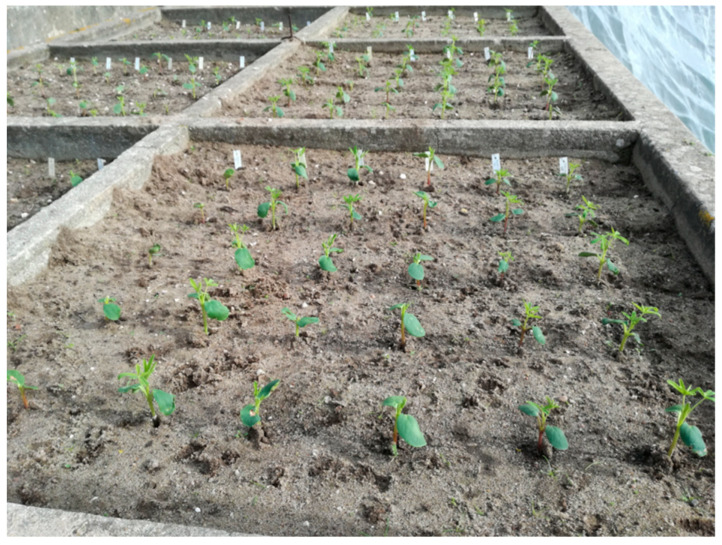
Experiment conduced in vegetation box for identifying *Lupinus mutabilis* accessions with anthocyanin in their organs. Each accession with 5 plants and 30 by vegetation box. From left to right: first, fourth and fifth rows with anthocyanin pigmentation and the three remaining row without anthocyanin pigmentation.

**Figure 4 plants-09-00583-f004:**
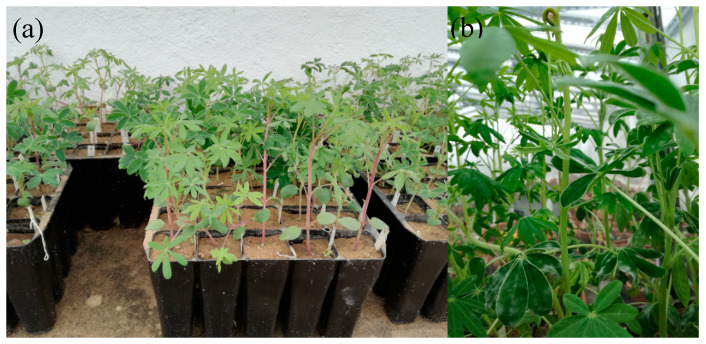
*Lupinus mutabilis* anthracnose resistance evaluation experiment. (**a**) plants with anthocyanin pigmentation and (**b**) without anthocyanin.

**Table 1 plants-09-00583-t001:** ANOVA results (95% confidence) performed on the severity data of the disease caused by *Colletotrichum lupini* on sixteen accessions.

Disease Score	Variable	Df	Sum Sq	Mean Sq	F Value	Pr (>F)
7 DAI ^1^	Experiment	1	7.444	7.444	1.4058	0.236388
	Accession	15	1438.188	95.879	18.1074	0.0
	Experiment:Accession	15	63.267	4.218	0.7966	0.681902
	Residuals	448	2372.178	5.295		
14 DAI	Experiment	1	4.561	4.561	0.9857	0.321331
	Accession	15	2795.750	186.383	40.2832	0.0
	Experiment:Accession	15	190.500	12.700	2.7449	0.000461
	Residuals	448	2072.816	4.627		
21 DAI	Experiment	1	30.061	30.061	6.577	0.010655
	Accession	15	2894.183	192.946	42.214	0.0
	Experiment:Accession	15	183.735	12.249	2.680	0.000630
	Residuals	448	2047.646	4.571		
AUDPC	Experiment	1	107.05	107.05	2.7789	0.096214
	Accession	15	20572.24	1371.48	35.6007	0.0
	Experiment:Accession	15	987.12	65.61	1.7030	0.047302
	Residuals	448	17258.79	38.5		

^1^ Days after inoculation.

**Table 2 plants-09-00583-t002:** Means and homogeneous groups resulting from the comparison test (Tukey, 95%) for the AUDPC of infection caused by *Colletotrichum lupini* on accessions of *Lupinus mutabilis* and *L. albus*.

Accessions	Species	Anthocyanin Presence ^1^	AUDPC	H.G. ^2^
Bunheiro Murtosa	*Lupinus albus*		0.79	a
LM34	*Lupinus mutabilis*	1	1.52	a
Prima	*Lupinus albus*		1.99	a
MISAK	*Lupinus albus*		2.28	a
LIB222	*Lupinus mutabilis*	1	2.35	a
I82	*Lupinus mutabilis*	1	2.52	a
XM-5	*Lupinus mutabilis*	1	3.12	a
LM13	*Lupinus mutabilis*	1	4.54	a
LM231	*Lupinus mutabilis*	1	5.45	a
Lublanc	*Lupinus albus*		11.96	b
LM18	*Lupinus mutabilis*	0 ^3^	13.27	b
Estoril	*Lupinus albus*		14.26	bc
XM1-39	*Lupinus mutabilis*	0	14.28	bc
Rio Maior	*Lupinus albus*		15.51	bc
Mutal	*Lupinus mutabilis*	0	19.16	c
LM268	*Lupinus mutabilis*	0	19.21	c

^1^ For *Lupinus mutabilis* accessions only; ^2^ Accessions with one or more letters in common do not differ in statistical terms for a significance level of 95%; ^3^ LM18 develops anthocyanin pigmentation in the areas surrounding anthracnoses.

**Table 3 plants-09-00583-t003:** ANOVA results (95% confidence) performed on the AUDPC of the infection caused by *Colletotrichum lupini* in ten *Lupinus mutabilis* accessions with or without anthocyanin pigmentation.

Variable	Df	Sum Sq	Mean Sq	F Value	Pr (>F)
Experiment	1	88.84	88.84	1.9712	0.161371
Anthocyanin content	1	12128.93	12128.93	269.1297	0.0
Experiment: Anthocyanin content	1	0.31	0.31	0.0070	0.933461
Residuals	296	13339.90	45.07		

**Table 4 plants-09-00583-t004:** *Lupinus mutabilis* accessions selected for use on anthracnose resistance experiment based on anthocyanin presence.

Accessions	Coat Seed Colour	Selection	Anthocyanin Presence	Organs with Anthocyanin
I82	White with brown crescent			
White with dark crescent and specks	Selected	Yes	Stem, petiole and leaflets
LIB222	White with specks	Selected	Yes	Stem, petiole and leaflets
LM13	White	Selected	Yes	Stem and petiole
LM18	Brown with over eye	Selected	No	
LM231	White with over eye			
White with crème crescent			
White with brown crescent	Selected	Yes	Stem and petiole
White marbled with over eye			
Brown with over eye and specks			
LM268	White with over eye	Selected	No	
Brown with over eye			
LM27	Brown with over eye and specks			
LM34	White with moustache with specks			
White marbled with over eye	Selected	Yes	Stem, petiole and leaflets
Brown with over eye and specks			
Dark			
Mutal	Brown with over eye	Selected	No	
XM-5	White			
White with over eye			
White with crème crescent			
White with dark crescent and specks	selected	Yes	Stem, petiole and leaflets
White with moustache with specks			
White marbled with over eye			
Dark			
XM1-39	White	Selected	No	
White with crème crescent

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
