# Peer review of "Response to Anthracnose in a Tarwi (Lupinus mutabilis) Collection Is Influenced by Anthocyanin Pigmentation"

_plants, 2020, doi:10.3390/plants9050583_

Round 1
Reviewer 1 Report
In References, page 13 line 405, reference 26 was not mentioned in manuscript. On the other side, on page 9 line 250 is mentioned reference 6. Is that refers to this 26 reference?
Author Response
In References, page 13 line 405, reference 26 was not mentioned in manuscript. On the other side, on page 9 line 250 is mentioned reference 6. Is that refers to this 26 reference?
R: Reference 26 is cited on page 7 line 220. About question 2 related with reference 6, In fact, citation 6 on page 9 line 259 seems to generate some confusion but it is really this citation. From citation 6, accessions of Lupinus mutabilis were selected in our collection for anthracnose resistance evaluation study. While citation 26 was used to design primers that amplify the ApnMat1 gene which is known to be more informative.
Reviewer 2 Report
In this manuscript, the authors state their notice that the anthracnose disease severity, cause by Colletothrichum lupini, was lower on anthocyanin-rich tarwi plants. Except this finding, nothing else is reported, either the correlation coefficient between the anthocycianin pigmentation and disease severity.
In my opinion the study seems preliminary and barely falls within the scopes of this journal. Therefore I think the submitted manuscript cannot justify publication.
Author Response
We acknowledge that the reviewer recognizes the interest of the results relating anthocyanin presence with less anthracnose severity. There are no specific comments or corrections to address.
Reviewer 3 Report
The manuscript aims to find genetic variability among L. mutabilis accessions towards anthracnose response. The results obtained suggest a correlation between the disease severity and the anthocyanin pigmentation of L. mutabilis. This is a useful contribution to the literature, particularly given that some accessions could be used as a source of resistance for future breeding program. The paper is well structured and the research carried out with appropriate scientific consistency. The experiments are congruent with the aims of the paper and the results are well reported. There are only few minor concerns to consider.
Line 39. Is there any literature to cite?
Line 83. The first result to describe is the obtaining of the strain NG001, the authors describe it in Materials and methods (line 202-203), writing it in results could be more appropriate.
Table 2. At LM18/LIB204 there is the note n.3, in the description below the table (line 128) the number is wrongly indicated (2 instead of 3).
Line 94-96. Does the anthocyanin pigmentation occur only in one case (as the authors have written fig.1c) or in all the experiments on stems of accession LM18/LIB204? The authors should better explain this point.
Line 146. Delete “performed by”.
Author Response
Line 39. Is there any literature to cite?
R: About citation on line 39, the first part of the introduction that comprises line 32 to 43 was based on the citation [1].
Line 83. The first result to describe is the obtaining of the strain NG001, the authors describe it in Materials and methods (line 202-203), writing it in results could be more appropriate.
R: Thank you for this observation, which in our view makes sense, we decided to put it in the methodology due to the need to standardize the names of the strains. However, to allow a better understanding of the text we made a small change in the text passing for: “Sequencing results of a 968-bp fragment of ApnMat1 gene for the two isolates (NG001, collected in Portugal, and RB221, collected in France, respectively with GenBank references MN783012 and MN783013) reveals 100% similarity between them.”
Table 2. At LM18/LIB204 there is the note n.3, in the description below the table (line 128) the number is wrongly indicated (2 instead of 3).
R: About note n.3, in fact it was a failure and we did the correction replacing the number 2 by 3 in the footnote.
Line 94-96. Does the anthocyanin pigmentation occur only in one case (as the authors have written fig.1c) or in all the experiments on stems of accession LM18/LIB204? The authors should better explain this point.
R: Thanks for this observation. We added “(consistently observed in all plants of this accession in both experiments)”.
Line 146. Delete “performed by”.
R: Thanks for this observation, the correction was made deleting performed by.